# Pig Pregnancies after Transfer of Allogeneic Embryos Show a Dysregulated Endometrial/Placental Cytokine Balance: A Novel Clue for Embryo Death?

**DOI:** 10.3390/biom10040554

**Published:** 2020-04-05

**Authors:** Cristina A. Martinez, Marie Rubér, Heriberto Rodriguez-Martinez, Manuel Alvarez-Rodriguez

**Affiliations:** Department of Clinical & Experimental Medicine (IKE), BHK/O&G Linköping University, SE-58185 Linköping, Sweden; marie.ruber@realgymnasiet.se (M.R.); heriberto.rodriguez-martinez@liu.se (H.R.-M.); manuel.alvarez-rodriguez@liu.se (M.A.-R.)

**Keywords:** cytokines, embryo transfer, maternal immune tolerance, embryo loss, endometrium, placenta, pig

## Abstract

Pig embryo transfer (ET) is burdened by high embryo mortality, with cytokines playing a significant role in recruitment of immune cells during embryo attachment and placentation. We hereby tested if their levels in endometrium and placenta from sows carrying hemi-allogeneic (artificially inseminated sows; C+ positive control) or allogeneic embryos (sows subjected to ET; ET) during peri-implantation (D18) or post-implantation (D24) are suitable mirrors of embryo rejection or tolerance after ET. Non-pregnant sows (C−) were used as negative controls. A set of cytokines was assayed in the tissues through multiplexed microsphere-based flow cytometry (Luminex xMAP, Millipore. USA). Fewer (58.7%. *p* < 0.003) conceptuses were recovered at D24 after ET compared to C+ (80.9%); with more than 20% of the ET conceptuses being developmentally delayed. Cytokine levels shifted during implantation. Anti-inflammatory IL-10 levels were significantly (*p* < 0.05) lower in ET sows compared to C+ at D24 of pregnancy. The C+ controls (carrying hemi-allogeneic embryos) consistently showed higher levels of pro-inflammatory TNF-α, IFN-γ, and IL-2 cytokines at D18 and IL-1α at D24, compared to the ET group. This clear dysregulation of pro- and anti-inflammatory cytokine levels in sows subjected to ET could be associated with an impaired maternal immune tolerance, explaining the high embryonic mortality of ET programs.

## 1. Introduction

Spontaneous conceptus loss in pigs resulting in a significant decrease in litter size is of major concern for its negative economic impact [1]. Almost 40% of the embryos produced under natural breeding or artificial insemination (AI) will not develop to term because of the conspicuous embryo mortality that occurs in this species, mainly during the peri-implantation period (Days 15 to 30) [2,3,4]. The situation is more dramatic after embryo transfer (ET), where approximately 70% of the embryos transferred to recipient females die in their reproductive tract [5,6]. Likewise, results from human, using allogeneic embryos from egg donation (ED) indicated maternal alterations compared to pregnancies after in vitro fertilization (IVF) of the mother’s own eggs (semi-allogeneic embryos), alterations related to an increased immune activity, comparable to the immunological rejection of a graft [7,8]. Although embryo mortality after pig ET has been a topic of increased interest, the contributing causes are still not well understood, particularly regarding the response of the maternal uterus to the stimuli produced by allogeneic embryos.

After hatching, the embryo initiates one of the most critical periods throughout pregnancy. The cross-talk on either side of the maternal-fetal interface is crucial to achieve endometrial receptivity during the peri-implantation period of pregnancy [9]. During this period, porcine conceptuses modulate the maternal environment toward successful pregnancy [10], being responsible of establishing endometrial receptivity, and a balance between pro- and anti-inflammatory signals [11] through the secretion of different molecules such as growth factors and cytokines [12,13].

Under physiological conditions, the embryo(s) is defined as hemi-allogeneic to the mother since it contains paternal genetic material, and therefore, produces proteins that are unknown to the immune system of the female. Although immune maternal recognition of the hemi-allogeneic embryo occurs, the maternal immune system is locally down-regulated, at least in human, which favors embryo implantation and the establishment of a state of maternal immune tolerance, while maintaining immune capacity against pathogens [14]. When embryos are allogeneic, e.g., containing both paternal and maternal material different from the recipient female, as it is the case during ET, the mechanisms regulating the maternal immune response to the embryo may be more complex and less efficient, jeopardizing the ability of many of these embryos to develop to term; a fact thoroughly recognized. Moreover, considering the interventional nature of the ET procedure, a maintained immune capacity against pathogens is most relevant, while preventing rejection of the transferred embryos [15,16]. Cytokines are a family of secreted immune modulators controlling the function and differentiation of both immune and non-immune cells [17,18]. During pregnancy, cytokines in the internal female genital tract are crucial for the establishment of a network of communication at the maternal-fetal interface; ultimately affecting the success of implantation, placentation, and the maternal immune tolerance to the allograft constituted by the embryos/fetuses and their extraembryonic membranes [19,20]. The immunological success of allograft tolerance depends upon the establishment of a balance between pro-and anti-inflammatory cytokines [21]. In this regard, T helper (Th) lymphocytes are the major producers of cytokines, and the balance between Th1 (pro) and Th2 (anti) cytokines defines pregnancy success [22]. Although the protective role of the Th2 cytokines as well as the harmful effects of Th1 cytokines has been demonstrated, other studies show that each cytokine follows specific patterns of expression, temporally, each day of the pregnancy [23]. These observations indicate that despite the beneficial or harmful effect of a cytokine during a specific moment of pregnancy, their determined up- or downregulation must be followed, a pattern that will decide pregnancy fate. 

This paper describes experimental work aimed to specifically investigate whether pro- and anti-inflammatory cytokine expression and/or their ratio in endometrial and placental samples during early pregnancy (Days 18 and 24) differ in response to the presence of hemi- or allogeneic embryos, testing the hypothesis that the condition of allogeneity is substantially involved in maternal embryo rejection after pig ET. 

## 2. Materials and Methods 

### 2.1. Experimental Design

In order to study the cytokine expression of uterine and placental samples after ET or AI, sows were assigned randomly to two groups; sows were either artificially inseminated (C+, n = 8), or subjected to surgical ETs (ET, n = 9) at Day 5 (D5; D0 = onset of estrus), with a pool of fresh embryos at the morulae stage (23 transferred embryos per recipient) (Figure 1). For achieving ETs, genetically unrelated sows were inseminated and used as embryo donors. Additionally, sows artificially inseminated with ejaculates containing dead spermatozoa were used as non-pregnant controls (C−, n = 8). Artificially inseminated sows (embryo donors, C+ and C−) received sperm doses from the same boar, in each of two replicates. 

To mimic the experimental conditions of the ETs, sham surgeries were also performed at D5 in C+ and C− sows. Endometrial and feto-placental samples were collected from all potential pregnant sows on D18 or D24 of pregnancy. 

On D18 or D24 of pregnancy, samples were collected and pooled from three different segments of the endometria (EN) and placentae (PL; Fetal-side). At D18 endometrial samples were collected only from the embryo attachment site (C+ and ET) because we were not able to distinguish non-implantation regions of the endometrium since embryo implantation was not fully achieved. At D24 endometrial samples were collected from both implantation (IPA) and non-implantation areas (NIPA) (C+ and ET groups). 

Each group had a N = 4 sows per period of evaluation (N = 12 at D18; N = 12 at D24; total N = 24). A set of cytokines (GM-CSF, IFN-γ, IL-1α, IL-1ra, IL-1β, IL-2, IL-4, IL-6, IL-8, IL-10, IL-12, IL-18, TNF-α, and TGF-β, 1, 2, and 3) and the ratio between TNF-α and IL-10 were analyzed in all experimental groups and conditions. 

### 2.2. Animals

All experimental procedures were assessed according to the 2010/63/EU EEC Directive for animal experiments and were revised and approved by the Ethical Committee for Experimentation with Animals of the University of Murcia, Spain (research code: 522/2019; 21 March 2019). Crossbred sows (Landrace × Large-White), parity 2 to 7, with a lactation period of 21–24 d were randomly selected at weaning and used for this experiment. All females were allocated into individual crates in a mechanically ventilated confinement facility at a commercial farm (Agropor SA, Murcia, Spain). The semen donor was a sexually mature boar (2 to 3 years of age) of proven fertility, housed in climate-controlled individual pens (20 to 25 °C) at a commercial breeding boar station for AI-dose semen production in Murcia (AIM Iberica, Spain). The animals had access to water ad libitum and were fed commercial diets according to their nutritional requirements.

### 2.3. Estrus Detection and Artificial Insemination

Detection of estrus was performed by snout-to-snout contact of females with vasectomized mature boars while applying back-pressure to test for standing estrous reflex twice daily, beginning one day after weaning. Only sows with a weaning-to-estrus interval of 4 days were selected for the experiment. Post-cervical inseminations were performed at 6 and 24 h after the onset of estrus. Sows were post-cervically inseminated (uterine body) 6 h and 24 h after the onset of estrus with 1.5 × 10^9^ dead (C−, negative controls) or alive spermatozoa (C+ and embryo donor sows), in 40 mL doses prepared with semen from adult boars extended with Beltsville Thawing Solution [24].

### 2.4. Embryo Recovery and Embryo Transfer

At Day 5 after inseminations, sows were sedated with azaperone i.m. (2 mg/kg body weight) and anesthetized with sodium thiopental (7 mg/kg body weight, i.v.) administered in the marginal vein of an ear. Isofluorane (3–5% in air) was used to maintain anesthesia and buprenorphine (0.3 mg/sow, i.m.) was employed as analgesic. After exposure of the genital tract, embryos were collected by flushing the tip of each uterine horn with 30 mL of Tyrode’s lactate-HEPES-polyvinyl alcohol medium [25] at 37 °C. Immediately after collection, the recovered embryos were evaluated under a stereomicroscope to assess embryo quality and developmental stage, according to the morphological criteria of the International Embryo Transfer Society [26]. Only embryos deemed of excellent quality were used for ETs. Immediately after evaluation, embryos (n = 23) were loaded into Tomcat catheters for their transfer into the recipients. The Tomcat catheter was loaded with air bubbles that separated the 30-μL drop of medium that contained the embryo from two drops of TL-HEPES-PVA before and after the embryo. One hour before the ET, each recipient received a single intramuscular injection of a long-acting amoxicillin suspension (Clamoxyl LA; Pfizer, Madrid, Spain) at a dose of 15 mg/kg. Embryos at the morulae stage (Day 5) were transferred to the tip of a uterine horn (5 to 6 cm from the utero-tubal junction) with a Tomcat catheter inserted through the uterine wall, which was previously punctured with a blunt Adson forceps [27]. Post transfer, all recipients were evaluated daily for behavioral changes, including signs of estrus beginning at 12 days post-ET until D18 and D24 when tissue samples from endometrium and placentae were collected.

### 2.5. Embryo and Corpora Lutea Evaluation

On D18 or D24 of pregnancy, embryos from ET and C+ groups were retrieved and evaluated for number, morphology, and state of development. Embryos with approximately half of the size of the normal ones and with less or no intra- or extra-embryonic vascularity were considered developmentally delayed. Also, number of corpora lutea was counted to calculate the percentage of recovered embryos/corpora lutea in C+ group. 

### 2.6. Tissue Collection and Preparation

On D18 or D24 of pregnancy, samples were collected from endometria and placentae (C+ and ET groups). In non-pregnant uteri (C− controls), endometria were collected from three mesometrial sites at random, while in pregnant uteri they were removed from the site of attachment (D18) or the site of implantation (D24), the non-implantation area (D24), as well as from the placentae (D18 and D24). For cytokine analyses, 10 mg of the endometrial or placenta samples were disrupted in Procarta lysis buffer (Affymetrix) containing 1 mM PMSF and 1X protease inhibitor cocktail (Sigma-Aldrich, Quimica S.A., Madrid, Spain) and centrifuged at 1500× *g* at 4 °C for 10 min. The supernatant was transferred to a new microfuge tube and stored at −80 °C until assayed for cytokine contents. 

### 2.7. Cytokine Analysis

Luminex^®^ xMAP^®^ technology, a multiplexed microsphere-based flow cytometric assay, was used to examine the presence and relative concentration of a battery of cytokines and chemokines including granulocyte macrophage colony-stimulating factor (GM-CSF), interferon-γ (IFN-γ), interleukin 1α (IL-1α), interleukin-1 receptor antagonist (IL-1ra), interleukin 1β (IL-1β), interleukin-2 (IL-2), interleukin-4 (IL-4), interleukin-6 (IL-6), interleukin-8 (IL-8), interleukin-10 (IL-10), interleukin-12 (IL-12), interleukin-18 (IL-18), tumor necrosis factor-α (TNF-α), and transforming growth factor β (TGF-β-1-3) in endometrial and placental biopsies. 

The porcine specific PCYTMG-23k-13PX and TGFβ-64K-03 commercial kits (Merck Chemicals and Life Science SA, Madrid, Spain) were used according to the protocols provided by the manufacturers. Briefly, for TGFβ-64K-03 kit preparation, samples were acidified (pH < 3) with 8 μL of 1.0 N HCl and diluted 1:30 with assay buffer to adjust to the range of the standard curve. A cytokine standard curve comprising six standard points was built for each cytokine with the highest standard point at 10,000 pg/mL and the lowest 9.8 pg/mL. Serum matrix, provided in the kit, was used to mimic the composition of the endometrial environment in the standard, control, and blank measurements. 

PCYTMG-23k-13PX kit preparation, samples were not acidified. A cytokine standard curve, comprising six standard points, was built for each cytokine with the highest standard point at 10,000 pg/mL and the lowest standard point at 3.2 pg/mL. Serum matrix, provided in the kits, was used to mimic the composition of the endometrial environment in the standard, control, and blank measurements. Two controls provided in both kits were added in singlets. Following sonication, bead solution was added to each well for incubation at 4 °C in the dark, for 18 hr. After incubation, the plate was emptied using a multiscreen vacuum manifold (Merck Millipore), washed twice, and the detection antibody was added and incubated at RT in darkness for 60 min before streptavidin–phycoerythrin addition and further incubated for 30 min. 

After washing, the plates were run on a Luminex 200 TM (Luminex Corp, Austin, TX, USA) with xPONENT soft-ware version 3.1.7 (Luminex Corp) for acquisition and Masterplex 2010 version 2.0.0.68 (Mirai BioGroup, San Francisco, CA, USA) for data analysis. The median fluorescent intensity was analyzed using a 5-parameter logistic curve-fitting to calculate the concentrations of the cytokines in the samples. The data of cytokine concentration are shown as mean ± SEM median and percentiles.

### 2.8. Protein Measurement

Total protein concentrations in all samples were estimated using a fluorometric assay: Qubit Protein assay kit (ThermoFisher, Waltham, MA, USA), where measurements were performed using the Qubit Fluorometer 4.0 (ThermoFisher, Massachusetts, USA). 

### 2.9. Statistical Analysis

Statistical analyses were carried out using the SPSS Statistics version 19 (IBM SPSS Statistics, Chicago, IL, USA). The percentage of recovered embryos in C+ and ET sows was obtained in relation to the number of corpora lutea counted on each ovary or in relation to the total number of transferred embryos, respectively. These percentages were compared using the Fisher’s exact test. Cytokine concentrations were normalized to total protein (expressed as a ratio of pg cytokine/mg protein). Normal distribution of data was tested using the Kolmogorov–Smirnov test. The variation in concentrations of cytokines among groups was analyzed two-by-two comparisons for two independent samples with the Mann–Whitney U-test. A *p*-value <0.05 was considered to be statistically significant.

## 3. Results

### 3.1. Reproductive Performance at Day 18 or 24 of Pregnancy after AI or ET

All C+ sows sampled at D18 or D24 were pregnant. All ET sows (n = 4) were pregnant at D18, and all but one (n = 4) of the sows appeared pregnant at D24 of pregnancy (80%). Despite this high pregnancy rate, the mean percentage of recovered embryos was significantly higher (80.9% vs. 58.7%; *p* < 0.003) in the C+ sows (accounted from the total number of ovarian corpora lutea than the ET group (accounted from the total number of transferred embryos), depicting a developmental delay in the ET group (Table 1; Figure 2); results that indicate that in ET pregnant recipients (allogeneic conceptuses), the embryo loss was 22.2 percentage points higher (*p* < 0.001) than C+ (hemi-allogeneic conceptuses) during the first month of pregnancy, and embryo developmental delay was more pronounced in ET group compared to the positive control (22.2% and 5.3%, respectively; *p* < 0.006).

### 3.2. Variations on the Levels of Cytokines During the Peri-Implantation Period of Pregnancy 

Levels of pro-and anti-inflammatory cytokines analyzed among groups at D18 and D24 of pregnancy are shown in Table 2. The expression of several cytokines was affected by the treatment (ET and AI), region of endometrium (IPA and NIPA), or day of collection (D18 and D24) (Table 3). Of particular interest were those cytokines which revealed statistically significant differences among groups. Those are hereafter presented in detail (Figure 3, Figure 4, Figure 5 and Figure 6).

#### 3.2.1. Pro-Inflammatory Cytokines

Effects of the treatment: TNF-α and IFN-γ concentrations were higher (*p* < 0.03) in IPA (IFN-γ) and placenta (TNF-α) from C+ sows compared to ET sows at D18 of pregnancy (Figure 3). Additionally, Figure 3 shows the levels of expression of pro-inflammatory cytokines found in ET placentas compared to the C+ placentas at D18 and D24 of pregnancy, respectively, where IL-2 was higher (*p* < 0.05) in C+ group compared to ET group at D18 and IL-1α and IL-1β were higher (*p* < 0.01 and *p* < 0.05. respectively) in C+ IPA compared to ET IPA at D24 of pregnancy. Of particular interest was the expression of IL-1β and IL-18 cytokines, which are closely related during the peri-implantation period of pregnancy. At D18, IL-1β and IL-18 were higher (*p* < 0.05 and *p* < 0.03, respectively) in C+ IPA compared to C− IPA (Figure 3). In the ET IPA, IL-1β showed a tendency to be higher in PL (*p* < 0.08) at D24 compared to the C+ PL (Figure 3).

Effects of the endometrial area: TNF-α was higher (*p* < 0.03) in ET IPA when compared to NIPA at D24 (Figure 4). Also, IL-1β was higher (*p* < 0.03) in ET IPA compared to ET NIPA at D24 (Figure 4).

Effects of the day of pregnancy: In the endometria from ET sows (IPA), TNF-α was higher (*p* < 0.03) at D18 compared to D24 of pregnancy (Figure 4).

#### 3.2.2. Anti-Inflammatory Cytokines

Effects of the treatment: The levels of TGF-β 1 and 2 were significantly different in C+ IPA compared to C− IPA at D18 (*p* < 0.03) of pregnancy (Figure 5). Additionally, significant differences were found in the concentrations of IL-10 among groups. While at D18 of pregnancy the levels of IL-10 in C+ sows (IPA) were significantly lower (*p* < 0.03) than in non-pregnant sows (Figure 5), this profile shifted toward increased IL-10 synthesis by D24 of pregnancy, where C+ showed a considerable increase (*p* < 0.03) of this cytokine compared to C+ at D18 (Figure 6). Interestingly, the ET levels of this cytokine were significantly lower (*p* < 0.03) when compared to C+ sows both in endometrium and placentae at D24 (Figure 5).

Effects of the day of pregnancy: TGF-β2 levels were higher (*p* < 0.05) in ET IPA at D24 over D18 of pregnancy (Figure 6).

### 3.3. The IL-10:TNF-α Ratio

In normal pregnancies, IL-10 and TNF-α expression seems to be antagonistic during the peri-implantation period. Table 2 shows the concentrations of these cytokines among groups at D18 and D24 of pregnancy. We observed that a IL-10:TNF-α ratio was significantly different (*p* < 0.01) at D24 compared to D18 in both C+ and ET groups (IPA) (Figure 7). Despite the non-significant difference found for this ratio between C+ (hemi-allogeneic) and ET (allogeneic) in both D18 or D24, we observed that it was 3 percentage points lower in ET group at D24 (1.21 and 0.89. respectively; Table 2 and Figure 7).

## 4. Discussion

Establishment and maintenance of pregnancy in the pig involves the activation of many molecular interactions between the developing conceptus and the maternal endometrium, leading to cellular rearrangements and the development of the dual origin of tissues building the pig placenta [28,29,30]. Estrogens secreted by the pig conceptus during the period of rapid elongation is the major signal for the maternal recognition of pregnancy [10,31] acting on the uterine lining and glandular epithelia to induce the secretion of uterine factors required for conceptus development, implantation, and subsequent pregnancy [17]. However, the embryo-maternal communication involves many other, less known, signaling molecules, directly originating from the presence of the embryo in the reproductive tract of the mother, which leads to morphological, biochemical, and immunological alterations of the uterine environment. These regulatory mechanisms may be more complex and less efficient in the case of ET pregnancies, where the transferred embryos are allogeneic, jeopardizing the ability of many of these embryos to develop to term. In agreement with this hypothesis, in the present study, the embryo loss at D24 of pregnancy was 22.2 percentage points higher in ET sows than after AI (C+).

Here, we studied the maternal response to the stimuli produced by allogeneic embryos. For that, we studied the differences of expression of pro- and anti-inflammatory cytokines in response to the presence of either hemi-allogeneic (after AI; C+) or allogeneic (after ET) embryos and also in their fetal placentae, during the peri-implantation (D18) and post-implantation (D24) periods of early pregnancy. We observed differences in the expression of many pro-inflammatory cytokines between groups and stage of pregnancy, including TNF-α, IFN-γ, IL-2, IL-8, IL1-α, IL1-β, IL-18, and IL-12. There is extensive evidence that during early pregnancy, the hemi-allogenic embryo is capable of inducing a response by the female immune system, to defend both mother and foreign conceptuses from pathogens while, at the same time, the maternal immune system is induced to tolerate the hemi-allogenic conceptus, despite carrying paternal antigens thus attaining a state of maternal tolerance [32,33,34,35]. A notable increment of endometrial pro-inflammatory cytokines occurs after estrogen signaling and maternal recognition [36]. Furthermore, maternal uterine proinflammatory responses occurring at this interval are regulated by inducible transcription factors such as nuclear factor kappa B (NFKB), which is stimulated by cytokines released by the elongating conceptuses within the uterus [37]. In the present study, the expression of pro-inflammatory cytokines (endometrial IL-2 and IFN-γ, and placental IL-2 and TNF-α), considered modulators of uterine receptivity, maternal immune tolerance, and vascular changes essential to nourish the developing embryos [38] was higher in C+ sows compared to ET during the peri-implantation period (D18). This paradox could be explained by the fact that in ET-allogeneic pregnancies, when the maternal immune system discriminates self- from non-self-antigens, there is an inappropriate fetal/maternal presentation, thus jeopardizing proper maternal immune tolerance and propitiating embryo rejection. Moreover, the shortage/lack of estrogen signaling released by the remaining allogeneic embryos by D18 of pregnancy might not be able to stimulate an appropriate inflammatory response in the endometrium. To that extent, the results have proven the hypothesis that allogeneity is of fundamental biological importance to embryo survival.

Additionally, at D18 of pregnancy, we found a clear increase of the pro-inflammatory interleukin IL-1β in the C+ group compared to non-pregnant sows. This pro-inflammatory interleukin and its signaling system are expressed in the porcine endometrium and conceptuses during the peri-implantation period [39,40,41] inducing increases of the MAPK and NFKB activities in the endometrium [42], which stimulates the expression of genes involved in immune modulation, cell adhesion, and angiogenesis [17,43]. Immediately following implantation, IL-1β expression rapidly declines and the uterine content of IL-1β slowly decreases over the next few days and returns to pre-elongation concentrations [39], which is in accordance with our results at D24 of pregnancy where endometrial IL-1β concentrations tended to decrease in AI-pregnant sows compared to the non-pregnant controls. Interestingly, IL-1β levels in the placenta showed a tendency to be higher in the ET group compared to the C+ group at D24 of pregnancy. This could be due to the fact that in porcine, conceptus estrogen declines after their elongation to initiate a second increase, sustained during the period of uterine attachment (days 18 to 25 of pregnancy) [44]. Since IL-1β is a potent stimulator of aromatase activity and of subsequent estrogen synthesis [43], we hereby hypothesize that placental IL-1β production is increased in ET sows in an attempt to counteract the lack of estrogen signaling from the non-rejected allogeneic embryos remaining in utero. Moreover, allogeneic placentae also showed a higher expression of IL-1α compared to C+ conceptuses at D24 of pregnancy, which has been shown to negatively impact cell growth [45], suggesting that the development of the allogeneic embryos may be delayed in comparison with hemi-allogeneic conceptuses. This fact evidently agrees with the morphological findings among the conceptuses analyzed after collection at D24 of pregnancy, whereas the hemi-allogeneic conceptuses were in a more advanced stage of development when compared to allogeneic; both the healthy conceptuses and the delayed counterparts. However, implantation and subsequent pregnancy success depends on the maintenance of a balance between pro- and anti-inflammatory cytokines, as we have previously stated, in order to tolerate the conceptus, and also to create an environment free of pathogens during pregnancy. In this sense, the widely studied transforming growth factor beta (TGF-β) was highly expressed in AI-pregnant sows at D18 (TGF-β1 and β2) of pregnancy compared to the non-pregnant sows, confirming its relevance for the attainment of the state of maternal immune tolerance [46]. The TGF-β family consists of structurally related anti-inflammatory cytokines with pivotal roles in angiogenesis, immunotolerance, embryogenesis, embryo attachment, and tissue remodeling [47]. A decreased expression of TGF-β has previously been reported to participate in maternal immune rejection of the embryo [48], by a lack of stimulation of trophoblast cells proliferation and attachment to the uterine wall [11]. Noteworthy, expression of both TGF-β2 was downregulated in the endometrium of ET+ sows at D18 over D24, suggesting they might be implicated in disruption of cell adhesion or as a sign of immune maternal rejection during the peri-implantation period. Furthermore, we observed the synthesis of the anti-inflammatory cytokines IL-4, IL-6, and IL-10 was upregulated in C+ (IL-4, IL-6, and IL-10) and ET (only IL-6) sows at D24 compared to D18. In general, the conceptus secretes immunomodulatory factors to moderate the inflammation and gradually switch the environment to an anti-inflammatory state to orchestrate an adequate placental growth and development of pregnancy [49,50,51]. The interleukin IL-10 seems to significantly participate in suppressing the pro-inflammatory environment in the maternal interface being of vital importance for normal pregnancy [52,53] by inducing trophoblastic cells to produce vascular endothelial growth factor C (VEGF C) and the aquaporin AQP1, which therefore stimulates placental angiogenesis [54], while low IL-10 levels has been associated with pregnancy complications [55]. In the present study, levels of expression of IL-10 were significantly lower in endometrium and placenta of ET sows when compared to C+ sows at D24, probably leading to a failure in suppressing the immune response, thus favoring allograft rejection. Furthermore, in normal pregnancies the increase of IL-10, the most potent suppressor of TNF-α synthesis, substantially decreases TNF-α levels, thus increasing the IL-10:TNF-α ratio to facilitate placentation and maintenance of pregnancy. Interestingly, pregnancy failure has been associated with this shift being disrupted i.e., no increase in IL-10 nor of the IL-10:TNF-α ratio [56].

After determining the individual concentrations for IL-10 and TNF-α during both periods of pregnancy in our study, we also evaluated whether the ratio of IL-10:TNF-α could provide better information on the balance of inflammation taking place during early pregnancy after ET. We found a clear dysregulation between these cytokines in ET sows at D24, when the anti-inflammatory IL-10 did not increase as it did in a regular AI-pregnancy. The IL-10:TNF-α ratio was similar between C+ and ET groups at D18 of pregnancy, but remained lower in the ET group at D24, possibly reflecting either an excessive inflammation or a disturbed local immune reaction, leading to the failure to switch the balance towards the expected anti-inflammatory environment, thus triggering embryo loss.

## 5. Conclusions

In conclusion, expression levels of specific pro- and anti-inflammatory cytokines differed significantly between sows carrying hemi-(AI; C+) or allogeneic (ET)- conceptuses in both endometria and placentae, showing a clear dysregulation of the inflammatory balance in the ET-group, which could be associated with maternal immune rejection of the allogeneic allograft. To this extent, the hypothesis tested was proven, therefore aid explaining the high embryonic mortality rates occurring during early pregnancy in ET programs.

## Figures and Tables

**Figure 1 biomolecules-10-00554-f001:**
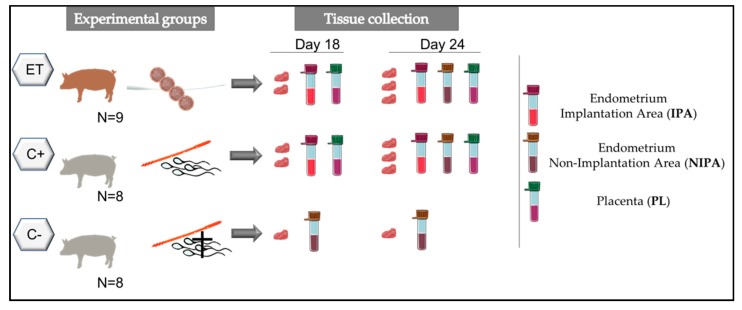
Schematic representation of the experimental design. Experimental groups: Sows subjected to surgical embryo transfer (ET), sows subjected to artificial insemination (C+), sows subjected to artificial insemination with dead sperm (C−). Tissue samples from different areas of the reproductive tract (Implantation area (IPA); non-implantation area (NIPA) or fetal placenta (PL) were retrieved after Day 18 or Day 24 of pregnancy.

**Figure 2 biomolecules-10-00554-f002:**
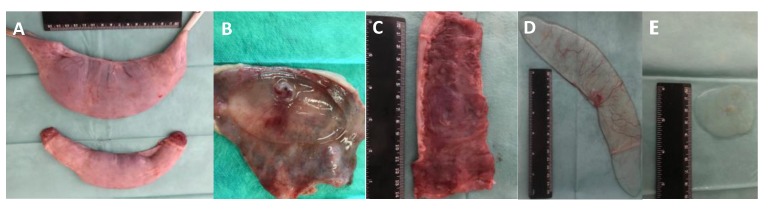
Representative images of two recipient uterine horns (**A**); normal and small chorionic sacs; and healthy (**B**,**D**) and arrested (**C**,**E**) pig conceptuses collected at D24 of pregnancy.

**Figure 3 biomolecules-10-00554-f003:**
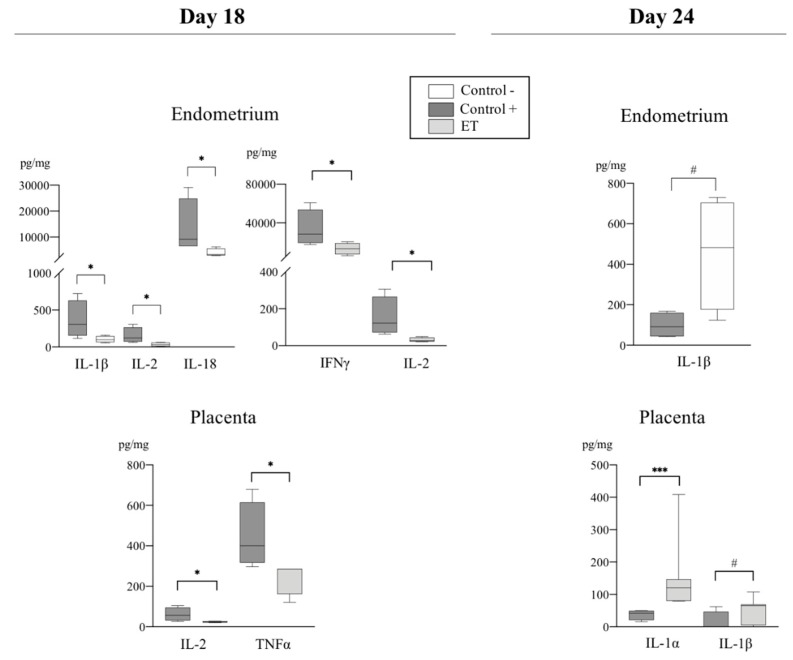
Relative abundance of pro-inflammatory cytokines (pg/mg of total protein) among experimental groups (C+, C−, and ET) at Day 18 and Day 24 of pregnancy in samples from endometrium and placenta. Data are presented by box plot showing median and inter-quartile range (Q1-Q3). Asterisks indicate significant differences among groups (* *p* < 0.05; *** *p* < 0.001; ^#^ Tendency *p* < 0.07).

**Figure 4 biomolecules-10-00554-f004:**
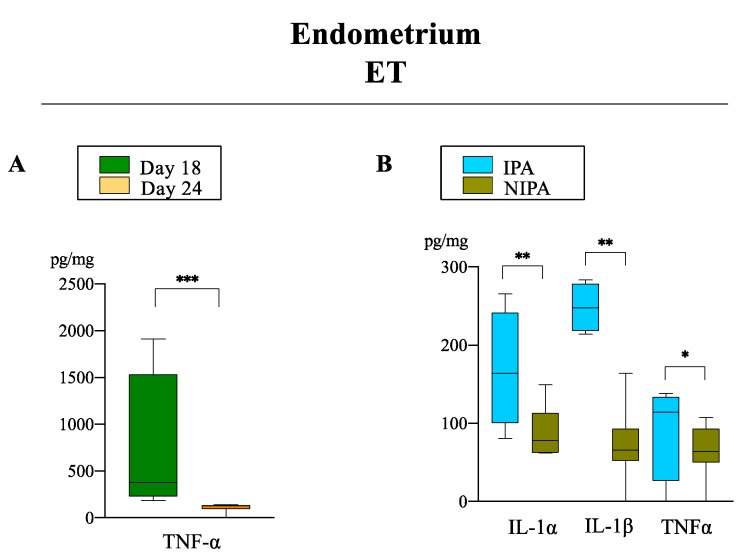
Relative abundance of pro-inflammatory cytokines (pg/mg of total protein) in ET group between days (**A**) and endometrial areas (**B**). Data are presented by box plot showing median and interquartile range (Q1–Q3). Asterisks indicate significant differences among groups (* *p* < 0.05; ** *p* < 0.01; *** *p* < 0.001).

**Figure 5 biomolecules-10-00554-f005:**
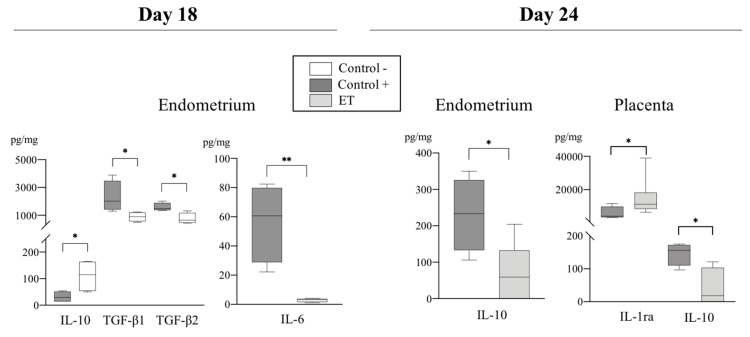
Relative abundance of anti-inflammatory cytokines (pg/mL) with different expression levels in endometrial and placental samples among experimental groups (C+, C− and ET) at Days 18 and 24. Data are presented by box plot showing median and interquartile range (Q1–Q3). Asterisks indicate significant differences among groups (**p* < 0.05; ** *p* < 0.01).

**Figure 6 biomolecules-10-00554-f006:**
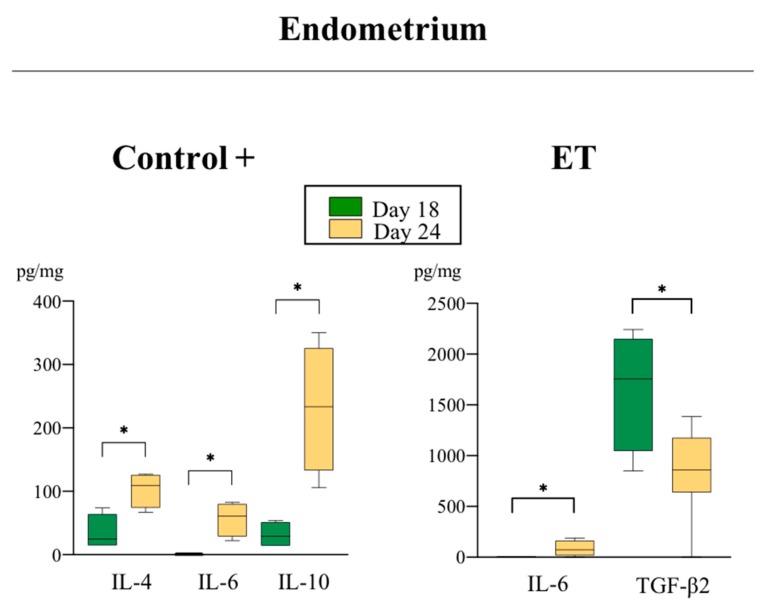
Relative abundance of anti-inflammatory cytokines (pg/mL) with different expression levels in endometrial samples (C+ and ET groups) in both periods of evaluation. Data are presented by box plot showing median and interquartile range (Q1–Q3). Asterisks indicate significant differences among groups (* *p* < 0.05).

**Figure 7 biomolecules-10-00554-f007:**
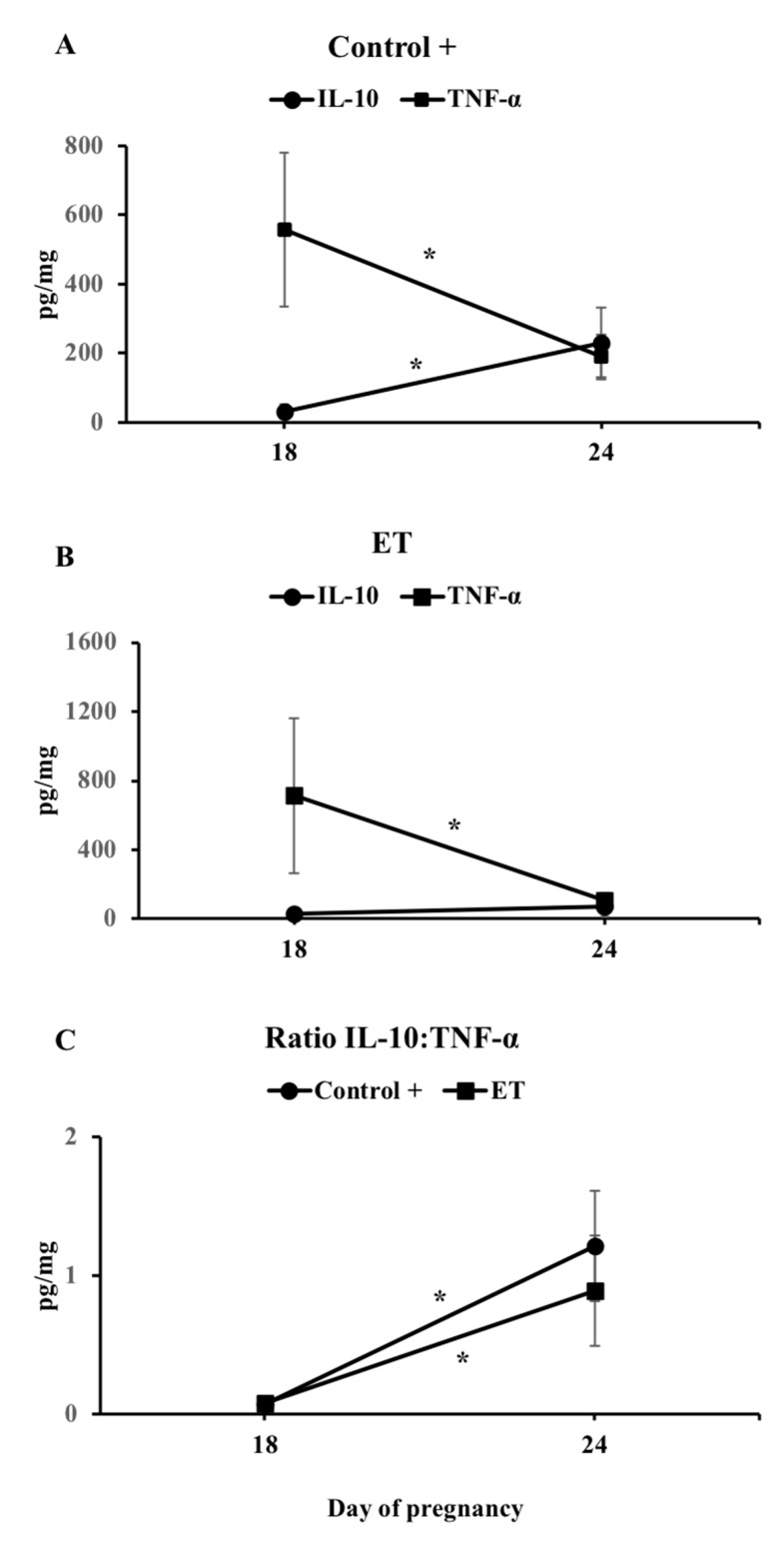
Levels of endometrial IL-10 and TNF-α in C+ (**A**) and ET+ (**B**) groups and IL10:TNF-α ratio (**C**) on D18 and D24 of pregnancy. The ratio was only significantly different (*p* < 0.01) when comparing D18 with D24 in both AI+ and ET+ groups. AI: sows artificially inseminated; ET: sows subjected to embryo transfer. Asterisks indicate significant differences among groups (*p* < 0.01).

**Table 1 biomolecules-10-00554-t001:** Pig embryos recovered at D24 of pregnancy in sows following artificial insemination (N = 4; C+) or embryo transfer of morulae (N = 4; ET).

Group	Corpora Lutea (n, mean ± sem)	Transferred Morulae/ET (n)	Total Recovered Embryos n (%) *	Normal Recovered Embryos n (%) *	Delayed Recovered Embryos n (%) **
**C+**	94 (23.5 ± 1.0)	-	76 (80.9) ^a^	72 (76.6) ^a^	4 (5.3) ^c^
**ET**	90 (22.5 ± 1.6)	92	54 (58.7) ^b^	42 (45.6) ^b^	12 (22.2) ^d^

* The percentage of recovered embryos in C+ and ET sows was obtained in relation to the number of corpora lutea counted on each ovary or in relation to the total number of transferred morulae embryos, respectively. ** The percentage of arrested embryos was calculated in relation to the number of total embryos recovered. Different superscripts in the same column indicate differences (^a^,^b^: *p* < 0.003; ^c^,^d^: *p* < 0.006; Fisher’s exact test).

**Table 2 biomolecules-10-00554-t002:** Levels (pg/mg of total protein) of GM-CSF, IFN-γ, IL-12, IL-18, IL-1α, IL-1β, IL-1RA, IL-2, IL-4, IL-6, IL-8, TGF-β1-3, IL-10, TNF-α, and IL-10:TNF-α ratio **i**n sows carrying hemi-allogeneic embryos (C+, n = 4), allogeneic embryos (ET, n = 4), and negative controls (not-pregnant, C−, n = 4) in samples of endometria (ET, C+: implantational areas) or NIPA (not-implantational areas) and fetal placenta (PL) at the peri-(D18) or post-implantation (D24) periods. Data are shown as Mean ± SEM.

	Day 18					Day 24						
CK	C−	C+	C+ PL	ET	ET PL	C−	C+ IPA	C+ PL	C+ NIPA	ET IPA	ET PL	ET NIPA
GM-CSF	40 ± 9.1	75.7 ± 19.2	52.3 ± 11.6	41.48 ± 13.5	42.1 ± 9.5	128.6 ± 74.5	47.2 ± 21.3	108.3 ± 13.1	113.0 ± 51	25.8 ± 6.2	84.1 ± 13.1	17.7± 5.7
IFN-γ	20,899.9 ± 5552.5	33,831.3 ± 9468.4	22,930.4 ± 5044.1	13,246.1 ± 2992.2	12,487.8 ± 2810	11,582.9 ± 2623.8	14,606.5 ± 634.5	21,030 ± 3019.4	30,452.2 ± 14,028.2	16,247.2 ± 5057.7	27,062.1 ± 7529.8	7820.2 ± 1636.2
IL-12	37.5 ± 30.7	23.3 ± 6.5	23.7 ± 4.5	29.9 ± 2.5	19.7 ± 2.5	187.4 ± 99.5	41.3 ± 15.7	0.0 ± 0.0	150.1 ± 101.5	28.3 ± 7.6	0.0 ± 0.0	30.3 ± 5.6
IL-18	3961.9 ± 778.3	13,493.9 ± 5363.5	6785.1 ± 482.5	11,497.2 ± 1106.5	8027.5 ± 1299.7	8914.1 ± 2895.4	7225.7 ± 2657.5	10,320.4 ± 2786.1	9367.8 ± 1691.2	8990.7 ± 2252	13,509.3 ± 3322.1	3403.2 ± 738.3
IL-1α	91.6 ± 20.2	113.2 ± 24.4	112.3 ± 26.5	147.2 ± 15.7	97 ± 31.2	98.05 ± 30.7	96.7 ± 17.9	37.4 ± 7.9	118.6 ± 38.8	168.6 ± 24.5	147.5 ± 38.5	88.7 ± 13.4
IL-1β	103.6 ± 21.7	363.3 ± 129.1	6170.5 ± 2716.6	253.5 ± 38.2	7674.7 ± 1759.4	454.3 ± 138.1	98.2 ± 31.7	15.6 ± 15.6	242.3 ± 156.7	202.4 ± 37.8	49.3 ± 13.6	73.9 ± 18.6
IL-1RA	719.5 ± 272.1	2042 ± 801.2	1749.2 ± 299.6	1575.6 ± 179.7	1682.7 ± 367.2	914.5 ± 95.3	2344.1 ± 647.4	5775.5 ± 1973.9	1714.2 ± 320.3	1279.8 ± 289.9	14,924.7 ± 3739.3	462.8 ± 135.5
IL-2	33.9 ± 13.5	153.2 ± 53.7	61.2 ± 16.6	30 ± 6.2	24.1 ± 1.6	519.7 ± 174.3	70.9 ± 15.1	268.3 ± 59.3	250.4 ± 169.5	30.2 ± 9.8	300.2 ± 83.1	29.1 ± 8.3
IL-4	49.7 ± 16.2	34.3 ± 13.9	20.9 ± 4.6	14 ± 3.8	20.4 ± 7.2	81.1 ± 49.3	103.1 ± 13.9	28.7 ± 1.7	333.2 ± 228.2	41.3 ± 18.3	19.5 ± 5.1	33.8 ± 12.2
IL-6	0.0 ± 0.0	0.0 ± 0.0	0.0 ± 0.0	0.0 ± 0.0	0.0 ± 0.0	0.0 ± 0.0	56.4 ± 13.3	40.9 ± 18.4	35.7 ± 20	85.8 ± 25.4	136.2 ± 34.2	11.7 ± 5.4
IL-8	0.8 ± 0.1	0.5 ± 0.1	0.7 ± 0.1	0.5 ± 0.04	0.6 ± 0.0	0.8 ± 0.8	1.1 ± 0.1	0.6 ± 0.1	0.7 ± 0.2	0.9 ± 0.06	0.6 ± 0.1	1.1 ± 0.1
TGF- β 1	888.3 ± 162.6	2300.4 ± 563.9	1739.5 ± 192.5	2092.4 ± 300.1	1995.1 ± 153.5	2062.3 ± 417.9	1358.2 ± 211.1	2936.9±	2848.5 ± 987.7	1387.5 ± 230.3	1856.1 ± 327.8	1208.1 ± 82.3
TGF- β 2	764 ± 195.6	1584.9 ± 147.6	2887.6 ± 688.1	1650.5 ± 294.1	3652.3 ± 552.1	1158.7 ± 228.5	1012.8 ± 237.3	1549.9 ± 397.4	1642.4 ± 631.9	842.8 ± 150.7	1343.9 ± 145.7	1217 ± 274.4
TGF- β 3	14.5 ± 5.1	21.4 ± 3.1	15.7 ± 3.3	22.8 ± 2.6	25.3 ± 7.4	16.9 ± 3.8	16.3 ± 2.5	15.9 ± 4.5	31.9 ± 10.8	18.1 ± 3.5	17.6 ± 3.4	22.6 ± 3.8
IL-10	110.9 ± 30.7	31.5 ± 10.1	62.0 ± 14.3	30.4 ± 5.4	76.3 ± 11.7	769.2 ± 359.6	230.6 ± 50.5	146.2 ± 17.3	608.6 ± 396.4	71.7 ± 16.1	41.2 ± 18.7	63.5 ± 24
TNF-α	375 ± 113.3	557.9 ± 178.4	444.1 ± 82.5	713.9 ± 402.2	244 ± 41.1	168.7 ± 27.2	189.8 ± 32.2	459.8 ± 85.7	305.3 ± 123.8	105.4 ± 26.6	548.9 ± 124.7	63.8 ± 13.3
IL-10:TNF-α (ratio)	0.4 ± 0.1	0.07 ± 0.02	0.15 ± 0.04	0.08 ± 0.02	0.36 ± 0.1	4.8 ± 2.6	1.21 ± 0.2	0.36 ± 0.09	1.58 ± 0.4	0.89 ± 0.2	0.2 ± 0.05	1.59 ± 0.2

Abbreviations: CK: Cytokines; C+: positive control group; C−: negative control group; ET: embryo transfer; PL: placenta; IPA: implantation area; NIPA: non-implantation area.

**Table 3 biomolecules-10-00554-t003:** Statistical differences (*p*-value, statistically different in bold) in cytokine levels among all groups and experimental conditions.

	Treatment-Day 18			Treatment-Day 24			Endometrial Area-Day 24		Day of Pregnancy	
CK	C+ vs. C− EN	C+ vs. ET EN	C+ vs. ET PL	C+ vs. C− EN	C+ vs. ET EN	C+ vs. ET PL	IPA vs. NIPA C+	IPA vs. NIPA ET	D18 vs. D24 C+	D18 vs. D24 ET
GM-CSF	0.149	0.248	0.564	0.248	0.234	0.126	0.248	0.561	0.149	0.308
IFN-γ	0.248	**0.043**	0.149	**0.021**	0.174	0.865	0.773	0.105	**0.021**	1
IL-12	0.386	0.564	0.386	0.149	0.496	1	0.248	0.908	0.386	0.865
IL-18	**0.021**	0.773.5	0.248	0.773	0.61	0.496	0.386	0.081	0.386	0.496
IL-1α	0.564	0.248	0.564	1	0.062	**0.007**	0.773	**0.01**	0.773	0.734
IL-1β	**0.043**	0.564.1	0.564	0.083	0.126	0.078	0.564	**0.024**	0.083	0.396
IL-1RA	0.083	0.773	0.564	0.149	0.174	**0.042**	0.386	**0.015**	0.773	0.308
IL-2	**0.043**	**0.021**	**0.043**	**0.021**	0.06	0.734	0.386	0.814	0.149	0.864
IL-4	0.386	0.248	0.773	0.248	**0.04**	0.307	0.564	0.765	**0.043**	0.494
IL-6	1	1	1	**0.014**	0.61	0.173	0.248	**0.045**	**0.014**	**0.03**
IL-8	0.386	0.248	0.248	0.248	0.126	0.497	0.083	**0.045**	0.083	**0.007**
TGF- β 1	**0.021**	0.773	0.564	0.083	0.497	0.174	0.083	0.2	0.083	0.126
TGF- β 2	**0.021**	0.564	0.248	0.564	0.61	0.497	0.564	0.183	0.149	**0.042**
TGF- β 3	0.248	0.773	0.248	1	0.497	0.734	0.149	0.563	0.149	0.234
IL-10	**0.043**	0.773	0.564	0.386	**0.026**	**0.016**	0.773	0.952	**0.021**	0.494
TNF-α	0.564	0.773	**0.021**	0.773	0.062	0.734	0.773	**0.024**	**0.021**	**0.007**
IL-10:TNF-α (ratio)	0.234	0.631	0.3	0.112	0.1	0.564	0.126	0.105	**0.002**	**0.009**

Abbreviations: CK: Cytokines; C+: positive control group; C−: negative control group; ET: embryo transfer; PL: placenta; IPA: implantation area; NIPA: non-implantation area.

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
