# Peer review of "Pig Pregnancies after Transfer of Allogeneic Embryos Show a Dysregulated Endometrial/Placental Cytokine Balance: A Novel Clue for Embryo Death?"

_biomolecules, 2020, doi:10.3390/biom10040554_

Round 1
Reviewer 1 Report
the article is well organize as well as the appropiate design of study, exlpaining the effect of allogenic transfer in sows, on inflammatory and anti-inflammatory pathway in D18 and D24 after pregnancy. the lost of embryos in allogenic transfer exlplain the jeopardiziation of the ET embryos. this is due to a different cross tolk between embryos and endometrium.
A good study well organized and very interesting for readers.
Author Response
We appreciate the referee’s comments.
Reviewer 2 Report
In the current work, authors studied the maternal response expression in terms of pro- and anti-inflammatory cytokines levels in sows with hemi-allogeneic (after AI) or allogeneic (after ET) embryos and the differences during the peri-implantation (D18) and post-implantation (D24) periods of early pregnancy as well as in different tissues (endometrium and placenta).
Results are difficult to follow in the present form. I suggest authors present them comparing the different effects studied: time (Day 18 vs 24), treatment (AI vs ET vs control) and areas (endometrium, no implantation tissue and placenta). Then statistical analysis should be performed to elucidate the effect of each variable. Figures should be useful to highlight significant differences but in a more organized form for clarity of the manuscript. The indication about pro-inflammatory or anti-inflammatory role of the cytokine would be useful for better understanding.
I have a concern about some of the current results because authors attributed embryo loss only of allogenic origin of transferred embryos but these embryos have been manipulated in vitro for a short time. Do authors think that this issue should not be taken into consideration? Viability of this kind of embryos could be lower due to the in vitro conditions even if they were few time outside the mother.
Detailed revision of each section can be seen below:
Introduction
Cytokine paragraph should be more extended explaining the differences between anti and pro inflammatory cytokines and the physiological changes during pregnancy.
Line 34: revise “…this species. mainly…”
Material and methods
-It should be recommendable a figure to illustrate experimental design. It is a bit confusing in the present form
In Table 2, in each row should be indicated the effect of time (Day 18 vs 24), the effect of the treatment (AI vs ET vs control), the effect of areas (implantation, no implantation and placenta) and the interaction between them. Statistical analysis by SAS program could be performed to analyse these data.
-How many replicates did you do?
-Could you mind explain what are you exactly referring with “sham surgery”?
-Change “pregnant females” by potential pregnant females
- Please, clarify the meaning of “Each group had a N= 4 sows per period of evaluation (total N=24)”
Line 89: Why Ethical Committee for Experimentation with Animals was approved by the University of Murcia? All the authors have Linköping University as filiation. Please, clarify
Line 104: how many boars did you used? Did you do a pool of semen for AI?
Line 120: Revise “…uterine wall. which was…”
Line 129: which part of the placenta? Maternal placenta? Can cytokines vary between different parts of placenta?
Line 130: eliminate (final concentration)
Line 131: please, indicate 14.000 r.p.m. in RCF unit
Line 141: analysis of PCYTMG-141 23k-13PX and TGFβ-64K-03 should be in a separate paragraph
Line 144: Molarity of HCl should be indicated
Line 144: what did you use for 1:30 dilution?
Line 165: recovered foetuses and count of corpora lutea were not described in material and methods before. Please, include them in 2.5 section. Also, the term of delayed embryos should be also referenced in that section and also in statistical analysis.
Line 174: change day 18 by Day 18
Results
Line 175-176: Please, clarify. In material and methods you indicated that ET females were 9 so this paragraph is unclear “All ET sows (n=4) were pregnant at D18, and all but one (n=4) of the sows appeared pregnant at D24 of pregnancy (80%)”. If are they 9 sows, total pregnancy rate should be 8/9=88.8%.
Line 177: For better understanding change sentence as follows: …..significantly higher (80.9% and 58.7%, P<0.003)…
Line 177: delayed embryos were not mentioned in the text
Line 183: change N by n throughout the table
Line 198: again numbers of animals are confusing. I guess that there were 4 animals in each moment (Day 18, n=4 and Day 24, n=4; total number of animals=8). Please, revise that notations coincide between Table legend and Table content. For example: NIPA is not in the table at Day 18.
In Table 2, in each row should be indicated the effect of time (Day 18 vs 24), the effect of the treatment (AI vs ET vs control) and the interaction between them
What is the differences between fig 4 and fig 2 and 3? In Fig 2 and 3 are representation of some cytokines Day 18 vs 24
Line 257-258: this sentence should be described in introduction paragraph and this ratio referred in experimental design
Fig 5: I suggest to name the figures as 5A, 5B and 5C. It is not clear the meaning of asterisk regarding the text. Please, revise the figure to illustrate the text and viceversa
Discussion
Line 302-307: Could you mind that supplementation with estrogen in preimplantation period to the mother could be useful to enhance maternal response?
Line 324-331: please, revise the paragraph. This is too long!
Line 345: Sentence should be reformulated for better clarity …. IL-6 and IL-10 was upregulated in AI+ and ET+ (only IL-6) sows at…
Line 359: In the result paragraph you indicated that IL10/ TNF-α ratio increased in both IA+ and ET+ groups and it was not different among groups. However, in discussion section you referred lower ratio in the ET+ group at D24. Please, clarify it
Author Response
In the current work, authors studied the maternal response expression in terms of pro- and anti-inflammatory cytokines levels in sows with hemi-allogeneic (after AI) or allogeneic (after ET) embryos and the differences during the peri-implantation (D18) and post-implantation (D24) periods of early pregnancy as well as in different tissues (endometrium and placenta).
Results are difficult to follow in the present form. I suggest authors present them comparing the different effects studied: time (Day 18 vs 24), treatment (AI vs ET vs control) and areas (endometrium, no implantation tissue and placenta). Then statistical analysis should be performed to elucidate the effect of each variable. Figures should be useful to highlight significant differences but in a more organized form for clarity of the manuscript. The indication about pro-inflammatory or anti-inflammatory role of the cytokine would be useful for better understanding.
Reply: We thank the reviewer for her/his suggestions. We have reorganized the text in the results section to allow a better understanding of the main findings of the paper, also, we have modified the figures aiming proper interpretation of the data. It is relevant to comment that in the text as well as in the figures only those significant differences between groups were represented. We have now clarified this fact (Lines 229-232 of the revised Ms). All conditions were subjected to a thorough statistical analysis as suggested.
I have a concern about some of the current results because authors attributed embryo loss only of allogenic origin of transferred embryos but these embryos have been manipulated in vitro for a short time. Do authors think that this issue should not be taken into consideration? Viability of this kind of embryos could be lower due to the in vitro conditions even if they were few time outside the mother.
Reply: We understand the concern of the reviewer and thank for the observation. In the control group(subjected to artificial inseminations, Control +) as well as in the negative control (Control -) animals were exposed to exactly the same conditions as the target group (ET), that is, surgeries in the control groups were performed on the same day of the embryo transfer. In CONTROL + group, embryos were retrieved, and re-transferred to the “MOTHERt” to mimic the experimental conditions of the ET group (materials and methods section; Line 88). This procedure ensured that the differences found in cytokine expression between treatments were exclusively related to the allogeneic condition of the embryos.
Detailed revision of each section can be seen below:
Introduction
Cytokine paragraph should be more extended explaining the differences between anti and pro inflammatory cytokines and the physiological changes during pregnancy.
Reply: The reviewer had a good point, thanks. The introduction section has been extended to accommodate differences in cytokine functions (Lines 64-72 of the revised Ms)
Line 34: revise “…this species. mainly…”
Reply: Done (line 33 of the revised Ms).
Material and methods
-It should be recommendable a figure to illustrate experimental design. It is a bit confusing in the present form
Reply: We have now included a schematic representation of the experimental design in the material and methods section (Fig. 1)
In Table 2, in each row should be indicated the effect of time (Day 18 vs 24), the effect of the treatment (AI vs ET vs control), the effect of areas (implantation, no implantation and placenta) and the interaction between them. Statistical analysis by SAS program could be performed to analyse these data.
Reply: After performing Kolmogorov-Smirnof test, we found that, our data were not normally distributed, also, we tried to transform the data in order to obtain a normal distribution, but the data were still not normally distributed. When your data is normally distributed, you can perform analysis of variance (multi-way tables) showing differences among groups within each factor and potential interactions among them. Since our data were not normally distributed and could not be transformed, we could only show results from non-parametric tests (comparing two groups within factor or checking within factor effects). The means and standard deviation of each variable in table 2 were reported to complete the information. However, we have now included a table in the results section (Table 3) showing the statistical variance (p-value) among all groups for each factor (Treatment, time and area)
-How many replicates did you do?
Reply: We performed 4 biological replicates per condition and 2 technical replicates for cytokine analysis.
-Could you mind explain what are you exactly referring with “sham surgery”?
Reply: We used the term “sham surgery” to refer to the fact that exact same surgeries were performed in the control groups to mimic experimental conditions of the target group (ET), to diminish/avoid confounding effects of surgery per se.
e.g. “In the sham surgery group, study participants undergo an imitiation of the surgical process going through everything that the actual surgery group does, from fasting to undergoing anesthesia to receiving surgical incisions—but they do not receive the procedure itself” (Advisory Board, 2017)
-Change “pregnant females” by potential pregnant females
Reply: Done (Line 89 of the revised Ms).
- Please, clarify the meaning of “Each group had a N= 4 sows per period of evaluation (total N=24)”
Reply: We retrieved tissue samples from 8 sows in the positive control group (AI+; now C+), 8 sows in the negative control group (AI-; now C-) and 8 sows in embryo transfer group (ET), total sows N=24. However, we retrieved the tissue samples at Day 18 (N=4 for each condition) and Day 24 (N=4 for each condition). We have now clarified these fact (Fig 1).
Line 89: Why Ethical Committee for Experimentation with Animals was approved by the University of Murcia? All the authors have Linköping University as filiation. Please, clarify.
Reply: Ethical permissions are to be applied for and obtained in the Country where the experimental animals are located, this applies to the entire European Union and National regulations of the member states. Thus, the approval was done by the University of Murcia, where the First author (CAM) was affiliated to before joining our research group at Linköping University as Post-doc.
Line 104: how many boars did you used? Did you do a pool of semen for AI?
Reply: We used sperm doses from the same boar for all AI’s (C+) to minimize biological variation.
Line 120: Revise “…uterine wall. which was…”
Reply: Done (Line 140 of the revised Ms).
Line 129: which part of the placenta? Maternal placenta? Can cytokines vary between different parts of placenta?
Reply: We retrieved samples from the fetal placenta, since pig is a non-invasive implantation species, it was easy to clearly collect just one side of the placenta (at least 3/6 layers of this epitheliochorial placenta type) in all sows and thus avoid confounding tissue-dependent cytokine discrepancies with endometrial samples not involved in the placenta (That was why we had analysed implantation (IPA) vs non-implantational (NIPA) areas. We have added this information in the material and method section. Line 92 of the revised Ms)
Line 130: eliminate (final concentration)
Reply: Done (Line 178 of the revised Ms)
Line 131: please, indicate 14.000 r.p.m. in RCF unit
Reply: Done (Line 163 of the revised Ms).
Line 141: analysis of PCYTMG-141 23k-13PX and TGFβ-64K-03 should be in a separate paragraph
Reply: Don (Lines 175-185).
Line 144: Molarity of HCl should be indicated.
Reply: Done (Line 176 of the revised Ms).
Line 144: what did you use for 1:30 dilution?
Reply: We used Assay Buffer included in the kit as indicated in the protocol. It has now been added to the text (Line 176).
Line 165: recovered foetuses and count of corpora lutea were not described in material and methods before. Please, include them in 2.5 section. Also, the term of delayed embryos should be also referenced in that section and also in statistical analysis.
Reply: A new section has been added to material and methods (2.5. Embryo and corpora lutea evaluation)
Line 174: change day 18 by Day 18
Reply: Done (Line 211 of the revised Ms).
Results
Line 175-176: Please, clarify. In material and methods you indicated that ET females were 9 so this paragraph is unclear “All ET sows (n=4) were pregnant at D18, and all but one (n=4) of the sows appeared pregnant at D24 of pregnancy (80%)”. If are they 9 sows, total pregnancy rate should be 8/9=88.8%.
Reply: We specified in the results section 3.1 that ALL (100%) of the sows were pregnant at D18 but 4 out of 5 (80%) were pregnant at D24 after ET. We agree with the reviewer that OVERALL the pregnancy rate in ET group reached 88.8%, but we found interesting to highlight the fact that the increase in mortality as achieved after D18 of pregnancy.
Line 177: For better understanding change sentence as follows: …..significantly higher (80.9% and 58.7%, P<0.003)…
Reply: Done (Line 214 of the revised Ms).
Line 177: delayed embryos were not mentioned in the text
Reply: It has now been added to the text (Lines 219-220 of the revised Ms).
Line 183: change N by n throughout the table
Reply: Done (Table 1).
Line 198: again numbers of animals are confusing. I guess that there were 4 animals in each moment (Day 18, n=4 and Day 24, n=4; total number of animals=8). Please, revise that notations coincide between Table legend and Table content. For example: NIPA is not in the table at Day 18.
Reply: Samples from non-implantation areas could not be retrieved at D18 due to the lack of full-embryo implantations. As mentioned in material and methods section (Line 93), at D18 we only collected samples from the site of attachment.
In Table 2, in each row should be indicated the effect of time (Day 18 vs 24), the effect of the treatment (AI vs ET vs control) and the interaction between them.
Reply: We have now added this information in Table 3.
What is the differences between fig 4 and fig 2 and 3? In Fig 2 and 3 are representation of some cytokines Day 18 vs 24
Reply: Figure 2 represented all cytokines modified at day 18, among groups. Figure 3 and 4 represented pro and anti-inflammatory cytokine changes among groups at day 24. We have modified the structure of the figures to facilitate the view of the results.
Line 257-258: this sentence should be described in introduction paragraph and this ratio referred in experimental design.
Reply: Done (Lines 74, 99-100 of the revised Ms).
Fig 5: I suggest to name the figures as 5A, 5B and 5C. It is not clear the meaning of asterisk regarding the text. Please, revise the figure to illustrate the text and viceversa
Reply: Done.
Discussion
Line 302-307: Could you mind that supplementation with estrogen in preimplantation period to the mother could be useful to enhance maternal response?
Reply: This suggestion is indeed very interesting since estrogen signaling is the key mechanism for maternal recognition in pigs, and could possibly aid in suppressing allograft rejection. However, manipulation of this kind is purely academic, since hormonal manipulation is not advisable in pig production and should be restricted. Further experiments, despite possibly difficult to pass an Ethical committee, would aid in deciphering estrogen roles.
Line 324-331: please, revise the paragraph. This is too long!
Reply: Done (Lines 476-482 of the revised Ms).
Line 345: Sentence should be reformulated for better clarity …. IL-6 and IL-10 was upregulated in AI+ and ET+ (only IL-6) sows at…
Reply: Done (Line 496 of the revised Ms).
Line 359: In the result paragraph you indicated that IL10/ TNF-α ratio increased in both IA+ and ET+ groups and it was not different among groups. However, in discussion section you referred lower ratio in the ET+ group at D24. Please, clarify it
Reply: In the results section we indicated that IL-10;TNF-α ratio was significantly different between D18 and D24 in both ET and AI groups, but it did not differ between ET and AI groups. We found, however, a non-significant decrease in this ratio in ET group compared to AI+.
Reviewer 3 Report
The study provides interesting data aimed to better understand the mechanisms of maternal embryo rejection or tolerance in pregnant sows. The topic is of scientific relevance and the study is well designed to test the proposed hypothesis.
Only minor corrections are suggested.
Check concordance between:
L 78-79
Sows from the three treatment groups were inseminated with sperm doses from the same boar
and L 93
The semen donors were sexually mature boars
and L 103-104
in 40 ml doses prepared with semen from adult boars
L 111-113
Indicate that only embryos of adequate/good quality were transferred
L 169-171
If a no parametric test was used why to check for normality?
L 181 (and L 282-283)
Indicate the actual difference in percentage points (22.2 vs "almost 30")
L 195-196
Check redaction and clarify what do consistent levels mean
L 329-330
no data are presented to support the asseveration "whereas the hemi-allogenic conceptuses were in a more advanced stage of development when compared to allogenic"; revise
Author Response
The study provides interesting data aimed to better understand the mechanisms of maternal embryo rejection or tolerance in pregnant sows. The topic is of scientific relevance and the study is well designed to test the proposed hypothesis.
Reply: We thank the reviewer for her/his comments.
Only minor corrections are suggested.
Check concordance between:
L 78-79
Sows from the three treatment groups were inseminated with sperm doses from the same boar
and L 93
The semen donors were sexually mature boars
and L 103-104
in 40 ml doses prepared with semen from adult boars
Reply: We have now clarified those sentences. However, even the AIs of the positive control sows were performed with semen doses from the same boar. We have to take into account that also mature boars were used for AI of the embryo donor sows (sows providing the embryos for allogeneic embryo transfer).
L 111-113
Indicate that only embryos of adequate/good quality were transferred
Reply: Done (Line 137 of the revised Ms).
L 169-171
If a no parametric test was used why to check for normality?
Reply: First we checked if the data had a normal distribution by using the Kolmogorov-Smirnov test. Then, due to the non-normality of the data, we used the described method in materials and methods section: “The variation in concentrations of cytokines among groups was analyzed two-by-two comparisons for two independent samples with the Mann–Whitney U-test. A P-value <0.05 was considered to be statistically significant”.
L 181 (and L 282-283)
Indicate the actual difference in percentage points (22.2 vs "almost 30")
Reply: Done (Lines 218 and 434 of the revised Ms).
L 195-196
Check redaction and clarify what do consistent levels mean
Reply: Done (Lines 233-236 of the revised Ms).
L 329-330
no data are presented to support the asseveration "whereas the hemi-allogenic conceptuses were in a more advanced stage of development when compared to allogenic"; revise
Reply: Correctly observed, thanks for the comment. We indeed performed a morphological evaluation of the embryos (see the new section 2.5). Differences in stage of development can be found in Table 1 of the revised version.
Round 2
Reviewer 2 Report
I would like to compliment authors on their excellent work of revision. Here are some minor revisions to made clearer your final version:
Line 94: fully instead sully?
Line 216: ….higher (P<0.001) than after C+… should higher (P<0.001) than C+
Line 269: add “at Days 18 and 24”
Fig 4: I suggest to name 4A and 4B and change the caption figure as follows: Relative abundance of pro-inflammatory cytokines (pg/mg of total protein) in ET group between days (Fig 4A) and implantation zones (Fig 4B).
Author Response
We thank the reviewer for her/his kind comments. We have made all the minor changes suggested.